# Assessing Intra-Bundle Impregnation in Partially Impregnated Glass Fiber-Reinforced Polypropylene Composites Using a 2D Extended-Field and Multimodal Imaging Approach

**DOI:** 10.3390/polym16152171

**Published:** 2024-07-30

**Authors:** Sujith Sidlipura, Abderrahmane Ayadi, Mylène Lagardère Deléglise

**Affiliations:** IMT Nord Europe, Institut Mines-Télécom, Univ. Lille, Centre for Materials and Processes, 59000 Lille, France

**Keywords:** compression molding, polymer matrix composites, thermoplastic resin, microstructural analysis, porosity, polarized light microscopy, fluorescence microscopy, scanning electron microscopy, multimodality

## Abstract

This study evaluates multimodal imaging for characterizing microstructures in partially impregnated thermoplastic matrix composites made of woven glass fiber and polypropylene. The research quantifies the impregnation degree of fiber bundles within composite plates manufactured through a simplified compression resin transfer molding process. For comparison, a reference plate was produced using compression molding of film stacks. An original surface polishing procedure was introduced to minimize surface defects while polishing partially impregnated samples. Extended-field 2D imaging techniques, including polarized light, fluorescence, and scanning electron microscopies, were used to generate images of the same microstructure at fiber-scale resolutions throughout the plate. Post-processing workflows at the macro-scale involved stitching, rigid registration, and pixel classification of FM and SEM images. Meso-scale workflows focused on 0°-oriented fiber bundles extracted from extended-field images to conduct quantitative analyses of glass fiber and porosity area fractions. A one-way ANOVA analysis confirmed the reliability of the statistical data within the 95% confidence interval. Porosity quantification based on the conducted multimodal approach indicated the sensitivity of the impregnation degree according to the layer distance from the pool of melted polypropylene in the context of simplified-CRTM. The findings underscore the potential of multimodal imaging for quantitative analysis in composite material production.

## 1. Introduction

Compression resin transfer molding (CRTM) is a process within liquid resin transfer molding to manufacture polymer matrix composites and is suitable for thermoplastic matrices, such as polypropylene (PP). For thermoplastic matrices, processing includes thermal regulation to bring the polymeric resin to its liquid state, which is then forced to flow through a fabric of reinforcing fibers, filling spaces between individual fibers at a micro-scale and between fiber bundles at a meso-scale. The impregnation vector during CRTM can be controlled by applying compressive mechanical pressure or a controlled displacement along the direction of the preform’s thickness (macro-scale). This pressure reduces unfilled spaces between the reinforcing filaments starting from the micro-scale and alters the preform’s permeability; at the same time, it promotes resin flow by creating streamlined pathways between and within bundles of reinforcing fibers. This antagonistic mechanism, where mechanical compression reduces permeability but facilitates filling of empty spaces, requires an in-depth understanding to effectively saturate the initially dry preform and reduce porosity (i.e., unfilled spaces and trapped air bubbles). For detailed insights into how physical forces interact with fluid dynamics to optimize resin distribution and quality in CRTM, refer to [1]. A significant challenge in CRTM processes is monitoring the flow of resin and the degree of impregnation, particularly in the thickness direction of a dry preform, known as the through thickness flow. This through thickness flow scenario is complex due to factors such as gravity, pressure gradients, permeability, temperature gradients, and capillary effects within the fabric’s microstructure. Simplifying assumptions can be considered where the through thickness flow can be simplified to the case where a fully saturated liquid zone (i.e., a pool of polymeric resin in the liquid state) transitions to unfilled, non-saturated fabric layers. In addition, when the saturated liquid zone is located below the preform, the combination of high-viscosity thermoplastic resins (compared to thermosets) and gravity allows for the neglect of capillary effects at the micro-scale of individual fibers.

Isothermal compaction can help to limit the presence of thermal gradients. The scenario involving pressure gradients and permeability variations is introduced by the authors in another study [2] and is referred to as simplified-CRTM. This displacement-controlled thermo-compression method (i) simulates CRTM’s impregnation phase, (ii) bypasses the injection stage of the molten polymeric resin, (iii) assumes isothermal conditions, and (iv) allows the resin to flow transversally through the preform’s thickness. Based on general considerations from the scientific literature about CRTM processes, to capture the anisotropic and non-linear behavior of resin flow, advanced computational fluid dynamics and experimental monitoring of the microstructure are needed to better understand these dynamics, ensuring the even saturation of the resin throughout the fibrous preform and mitigating potential residual porosity. Due to the complexity of real-time monitoring of fluid dynamics at multiple scales, post-manufacturing characterization of the microstructure can help to assess the porosity and degree of impregnation as a first step toward more complex analyses, passing through intermediate state impregnation levels. A literature survey focused on microstructure characterization of polymer matrix composites using 2D imaging techniques is provided in Table 1. This survey highlights studies on polymer matrix composite materials, including carbon fiber-reinforced polymers (CFRPs) and glass fiber-reinforced polymers (GFRPs), utilizing 2D imaging techniques such as optical microscopy (OM) and scanning electron microscopy (SEM). The analysis indicates 2D imaging techniques are mainly destructive, involving surface preparation and sample cutting to conduct subsurface observations and provide information about the microstructure in the form of multidimensional spatial arrays known as “images”. Authors frequently associate both destructive and non-destructive imaging methods to analyze microstructure-related aspects, including residual porosity, morphological aspects of fiber bundles or porosity, and impregnation. For porosity characterization, Purslow [3] utilized OM and SEM to assess porosity in CFRPs at the micro-scale, focusing on quantification and distribution, while Liu et al. [4] characterized porosity shape, size, and location at the meso- and micro-scales, linking these observations to processing parameters. Abdelal et al. [5] compared methods for characterizing porosity in glass fiber-reinforced composites (GFRCs), primarily using OM and µCT (micro-computed tomography). Gagani et al. [6] used OM to determine porosity location and quantification in GFRPs, and Ekoi et al. [7] investigated fatigue-induced damage in 3D-printed CFRPs using SEM. Zou et al. [8] examined residual porosity in aramid fiber-reinforced polymers using SEM and µCT. Regarding bundle morphology, Kabachi et al. [9] performed image-based characterization at the macro- and meso-scales using OM, while Breister et al. [10] studied bundle interactions in vinyl ester polymer using OM and SEM. Liu et al. [11] investigated high fiber volume fraction thermoplastics using SEM. For impregnation and resin flow, Ishida et al. [12] used OM during the compression process of thermoplastic composites, capturing images at different holding times to track impregnation and flow front progression at the meso- and micro-scales. Little et al. [13] analyzed oven-cured CFRP samples using density measurement, OM, environmental SEM, and µCT, while Eliasson et al. [14] used OM with neural network-based segmentation for void characterization in CFRP laminates. This non-extensive literature review indicates that various 2D imaging techniques are effective for both the qualitative and quantitative inspection of polymer matrix composites. However, several gaps remain unaddressed or insufficiently documented: (i) Studies lack comparative analyses of the same region of interest (ROI) under different imaging techniques. Indeed, multimodal imaging approaches, commonly used in the biological and medical fields, combine information from different images obtained through multiple techniques to create a richer and more accurate synthetic 2D image, overcoming the limitations of individual techniques [15,16,17,18,19,20,21]. The use of multimodal imaging in the context of polymer matrix composites is poorly documented (refer to Appendix A for a brief literature review exploring keyword co-occurrence relationships [22], and graphical illustrations created using the open access software VOSviewer, version 1.6.20 [23]). (ii) The analysis of partially impregnated composite samples using 2D destructive imaging techniques requires extensive surface preparation, such as polishing, yet the quality of these prepared surfaces is poorly documented. (iii) The observation of the entire thickness of composite samples is often needed, but there is a lack of consideration regarding the trade-off between the field of view, the smallest detectable detail, and the achievable resolution. (iv) Abdelal et al. [5] compared methods for characterizing porosity in GFRCs, employing ultrasound, burn-off tests, serial sectioning with OM, and µCT. They used a fluorescent dye mixed with epoxy as a resin mount and with talc powder on polished surfaces to highlight voids, although fluorescence microscopy (FM) was not used due to the lack of a UV source.

The current research explores the potential of multimodal 2D imaging techniques for qualitative and quantitative microstructure analyses of glass fiber-reinforced polypropylene matrix composites manufactured using isothermal compression molding to generate partially impregnated composite plates. A first manufacturing configuration based on film stacking is used to generate a reference composite plate formed of alternating thermoplastic films and woven unidirectional glass fiber plies. A second configuration based on simplified-CRTM is then used to manufacture two other plates by varying the compaction ratio during displacement-controlled impregnation. The aim is to generate composite plates with different impregnation levels to examine how the polypropylene (PP) matrix saturates a preform of six layers of woven glass fiber that are stacked as [0/90]_3_, aiming to understand resin distribution and localize porosity. The study addresses the potential of multimodal imaging (i.e., using different microscopy techniques) for analyzing the same composite microstructure for two main objectives: (i) quantifying porosity (i.e., unfilled spaces within the polymeric matrix) and (ii) assessing the degree of impregnation (i.e., saturation degree of intra-bundle spaces post-manufacturing) at a meso-scale of cross-sections of glass fiber bundles. The targeted techniques include optical microscopy using polarized light (PLM) for surface quality inspections and multimodality based on fluorescence microscopy (FM) and scanning electron microscopy (SEM) for quantitative analyses. The integration of these techniques provides a comprehensive assessment of the composites’ internal structure, contributing to the optimization of other manufacturing processes such as CRTM, including through thickness flow-dominated impregnation scenarios.

## 2. Materials and Methods

### 2.1. Materials

The matrix material employed was a commercial-grade polypropylene (PP) thermoplastic, identified as PPC13442 by Total^®^ (France), featuring a density of 0.905 g/cm^3^, a melting point of 165 °C, and a melt flow index of 100 g per 10 min. This PP was provided in form of pellets and was processed into thin films through thermocompression, yielding films with a mean thickness of 0.57 mm (±0.03 mm). For reinforcement, an experimental unidirectional (UD) woven glass fiber (GF) provided by Chomarat^®^ (France), referenced as JB111, were utilized. This GF exhibited a density of 2.55 g/cm^3^ and an areal density of 1054 g/m^2^. Rectangular UD plies measuring 375 × 375 mm^2^ were manually sectioned from larger length-scale rolled sheets. Rectangular composite plates with the dimensions of 375 × 375 mm^2^ were fabricated, each incorporating six of UD woven glass fiber plies arranged in a [0/90]_3_ sequence and seven thin sheets of PP matrix.

### 2.2. Methods

#### 2.2.1. Fabrication of Composite Plates

The goal of this section is to manufacture glass fiber-reinforced polymer composites (GFRPs) as partially impregnated plates, each with a different level of PP matrix saturation. As a reminder, the aim of the current research study is to evaluate multimodal imaging for assessing porosity and not optimizing the manufacturing of composite plates. In this context, three composite plates with varying impregnation levels were fabricated using isothermal compaction molding using an industrial-scale press (Pinette PEI^®^, France) capable of 120 tons of controlled force and precise top mold displacement to within ±0.1 mm (Figure 1a). During manufacturing, the press executed programmable cycles for temperature, displacement, and force, ensuring controlled manufacturing conditions. The glass fiber-reinforced polypropylene composite plates were produced using the same mold schematically illustrated in Figure 1b and which is equipped with a venting part allowing the evacuation of excessive polymer. The targeted theoretical heights of the composite plates were modified while maintaining the exact same constituents (i.e., seven PP films and six GF plies), as shown in Figure 1c,d, and can be verified from the column indicating the initial masses before manufacturing in Table 2. A first plate was manufactured using a film stacking configuration (Figure 1c), alternating glass fiber (GF) plies and PP films. A fixed compression force of 21 kN was applied to the top part of the mold (Figure 1b), and the mold’s cavity-height evolution during the process was recorded (Figure 1e). At this imposed level of compressive force, the preform was barely compacted (compared to the other plates within this same study), and impregnation was primarily due to the fixed compression force. Film stacking was selected to limit the compression level and maintain the fiber bundles undisturbed, based on force-driven control of the compression process. As the polymer melted, the mold gap was expected to progressively decrease due to the force-controlled process. The final thickness of the obtained plate was considered as the reference height for the other composite plates and for assessment of the compaction ratio (*Cr*) according to Equation (1):(1)$$Cr=\frac{h_{ref}-h_{final}}{h_{ref}}$$
where $h_{ref}$ is height of the plate manufactured using configuration 1 and $h_{final}$ is height of consecutive plates manufactured using configuration 2.

The other two plates were manufactured using isothermal compression molding similar to simplified-CRTM by positioning all solid-state PP sheets below the stack of GF plies (Figure 1d). Impregnation levels were controlled through a displacement-controlled process, targeting the compaction ratio of the preform as in a previous study by the authors [24]. The heating configuration was similar to film stacking, with isothermal compactions conducted at 215 (±2) °C. As the PP melted, a one-sided fluid pool formed below the preform. Due to the high viscosity of PP (compared to thermoset resins), capillary-driven flow into the preform gaps was restricted, making mechanical pressure during compression the primary driver for the impregnation of the GF bundles. Variations in the impregnation quality were influenced by the balance between the preform’s mechanical deformability and reduced permeability under compression, potentially resulting in residual porosity, including dry zones and trapped air bubbles. Some porosity may also result from PP shrinkage post-cooling [25]. No distinction is made between the causes of residual porosity in the context of the current study. This procedure is denoted by simplified-CRTM through the following sections. The simplified-CRTM procedure involved a ten-segment program, which included temperature-regulated heating, displacement-controlled compression, and cooling steps. Initially, two consecutive heating segments were carried out, with conductive heating using calorific oil to stabilize the mold temperature at 100 °C. This was followed by electric heating to achieve 215 (±2) °C. Once the target temperature was reached, six displacement-controlled segments, each lasting 300 s, were automatically activated to conduct staged compaction, progressively impregnating the reinforcement and achieving the targeted final thickness. The final two segments focused on cooling the plates while maintaining the final mold height, reducing the temperature to around 40 °C before opening the mold and extracting the plate. This displacement-controlled manufacturing configuration ensured the control of the final thicknesses of the plates, and thus, control over compaction ratios, which help in varying the impregnation levels. The input (i.e., displacement) and output (i.e., force) of the manufacturing process for the manufactured plates using simplified-CRTM are detailed in Figure 1f,g.

The final thickness control of the manufactured plates was achieved through metrological inspections, providing global average thicknesses from at least five locations on each manufactured plate. Additionally, optical microscopy inspections, including at least ten control points, were conducted on one localized sample extracted from the center of each composite plate (Figure 2a). Based on these microscopy inspections, compaction ratios of 0%, 30%, and 41% were obtained using Equation (1), as detailed in Table 2. A limited difference between the global metrological and local microscopy inspections was observed, confirming a relative uniformity in the thickness of the manufactured composite plates. Since the imaging techniques in the following sections will focus on these same samples used for optical microscopy-based height inspection, the corresponding compaction ratios were used to establish a simplified nomenclature for the composite plates: Cr_0%, Cr_30%, and Cr_41%, as denoted in Table 2. This nomenclature will be consistently used to reference the plates in subsequent sections. The final fiber volume fractions (V_f_) of all three composite plates post-manufacturing, compared to the target volume fractions (V_f_*) set before manufacturing, are reported in Table 2. Additional burn-off tests, according to ASTM D 2584, were conducted similarly to those reported in another study by the authors of [24] to quantify the V_f_ within the manufactured thermoplastic composite plates. As provided in Table 2, the obtained fiber volume fractions were 38.6%, 58.4%, and 64.9%, which were within same range of microscopy- and metrology-based evaluations. For each of the manufactured plates, a certain amount of polypropylene (PP) escaped the mold’s cavity, with weight losses of 3.4%, 23.2%, and 25.5% for the Cr_0%, Cr_30%, and Cr_41% plates, respectively. The significant losses observed in plates manufactured using the simplified-CRTM configuration can be explained by the unconstrained boundaries of the preform within the mold’s cavity, allowing in-plane polymer squeeze flows at the edges. As this study focuses on the microstructure inspection of partially impregnated composite plates using multimodal imaging, and not on correlating microstructure with manufacturing parameters, the limited constraints applied to the edges of the preforms during isothermal compression molding were considered beyond the scope of the current study.

#### 2.2.2. Mechanical Polishing of Partially Impregnated Composite Samples

After manufacturing the composite plates, one sample of 10 × 20 mm^2^ from the central zone of each composite plate, as illustrated in Figure 2a, was cut using a water-lubricated diamond saw and dried for 12 h at 40 °C to eliminate residual water. The surface preparation protocol for these dried samples relied on resin embedding and mechanical polishing to expose the subsurface within the microstructure. The mechanical polishing plane was considered perpendicular to the 0°-oriented bundles (in layers 1, 3, and 5) and tangential to the 90°-oriented bundles (in layers 2, 4, and 6). Mechanical polishing is mainly used for well-impregnated glass fiber-reinforced polypropylene composites to achieve a glossy mirror surface with controlled surface roughness [26,27]. However, in the presence of residual porosity and unsaturated regions within the preform where glass fibers are not fully impregnated by the polymeric resin, mechanical damage to unstable fibers limits surface quality control. Indeed, during polishing operations, glass fibers are subjected to quasi-static compressive forces (from the sample holder) and dynamic shear forces (from the rotating polishing discs), causing the breakage of single fibers and debonding of poorly impregnated fibers (Figure 2). To the best of the authors’ knowledge, no study has explicitly addressed the challenge of polishing partially impregnated thermoplastic composites. To address the specific challenge of polishing samples extracted from partially impregnated composite materials, this section introduces a four-step surface preparation protocol, which is illustrated in Figure 3. The aims are: (i) minimizing glass fiber breakage, particularly within fiber bundles perpendicular to the polishing surface (i.e., oriented at 0°), and (ii) helping the detection of process-induced residual porosity during image-based inspection after surface preparation while minimizing surface damage within these bundles. Surface control during the surface preparation protocol was principally qualitative, and quantitative inspections such as surface roughness control were considered beyond the scope of the current study. The first step involved a non-classic resin embedding operation of the cut and dried samples, by applying a fluorescent dye-enriched resin mount. EpoFix (Struers^®^, Denmark) resin and EpoDye (Struers^®^, Denmark) fluorescent agent were mixed respecting a ratio of 5 g dye per 1000 mL resin before adding the hardener with respect of a volume mixing ratio of 15 to 2. The mixture was degassed in a vacuum chamber for 20 min and then immediately poured into a circular mold of approximately 30 mm of inner diameter and 20 mm depth containing a carefully oriented and clamped composite sample. The mixture was then left to cure at room temperature for 12 h to ensure full polymerization. The second step involved height-controlled serial polishing using abrasive discs of 500, 1200, and 4000 grits and an automated polishing machine (TegraMin 20, Struers^®^, Denmark). This equipment allows the control of parameters including the compression force applied on the sample, rotational speeds of the sample and the disc, and the duration of each polishing step.

The inspections visually checked the 0°-oriented bundles where the limited presence of surface scratches and relatively good circularity of single fibers are signs of acceptable results at 4000 grit. In the case of the 90°-oriented bundles, as the polishing exposes new surfaces over time, their quality was not considered for surface control during this second step of the protocol, where a significant amount of broken and debonded single fibers are observed. These broken fibers in vicinity of 90°-oriented bundles caused the deterioration of the surface quality, notably deep scratches within polymer-rich zones and in 0°-oriented bundles. An illustrative example of surface state progression during polishing and following grit changes is provided in Figure 2b. The 500 grit, which is considered the roughest, was considered to remove about 0.3 mm/min from the composite sample to be away from any potential structural damage caused during the sample cutting operation. Polishing parameters for each grit level were selected through an extensive trial-and-error procedure combined with polarized light microscopy inspections of the surface using a microscope equipped with a digital camera.

After finishing the polishing at 4000 grit and qualitatively inspecting the quality of the 0°-oriented bundles, a third step of the surface preparation procedure was applied. It consisted of a second embedding of the exposed surface using the fluorescent dye-enriched epoxy mixture in the same proportions as in the first step. Precautions were taken to add an around 5 mm thick layer without changing the outer boundaries (i.e., bottom surface and cylindrical side wall) of the existing resin mount. The purposes of these steps are to (i) mechanically constrain all exposed and poorly impregnated glass fibers and (ii) seal new exposed porosity (i.e., including dry zones, trapped air bubble cavities) by the conducted sequential polishing steps and potential surface damage, such as scratches and localized fiber breakage. The fourth step of the surface preparation procedure consists of repolishing, assisted with the metrological control of the heights at a precision level of ±0.005 mm, to re-expose the same surface reached by the end of the second step using 500-, 1200-, and 4000-grit discs. This step removes the excess resin from the second embedding (i.e., the third step) while preserving the same microstructure exposed at 4000 grit by the end of the first polishing. The control of this operation is based on comparisons with the reference height of the resin mount after the first polishing step at 4000 grit. Additionally, microscopy inspections were simultaneously conducted on 0°-oriented bundles while ensuring that the fluorescence-enriched resin maintains transversely oriented single fibers within 90° bundles, limiting their debonding. For the glass fibers within the 0° fiber bundles, the interstitial spaces from dry zones or localized surface micro-cracks caused by the polishing from step 2 were expected to be infiltrated by the fluorescence-enriched resin, thus limiting further damage to fiber circularity and breakage. The polishing operation at 4000 grit was stopped once the target surface was fully exposed with a distance control of ±0.005 mm. Two finishing steps, using diamond particles of 6 µm and 3 µm, were then applied to refine surface quality and enhance the circularity of fibers, particularly in the 0°-oriented bundles (Figure 2b). It is worth noting that the sequence of polishing parameters (force, rotation speeds, and durations) used in this study cannot be considered unique or optimal, but it is considered reliable enough to guarantee similarity of surface states with minimal fiber breakage within the 0°-oriented bundles and minimal fiber debonding from the 90°-oriented bundles when applied to the considered partially impregnated samples. Additionally, all steps of the polishing procedure included ultrasonic cleaning in ethanol for the removal of material residues and surface cleaning. The main qualitative and quantitative focus will be on layers where bundles are oriented at 0°. For the layers where fiber bundles are oriented at 90°, the fluorescence-enriched resin is expected to saturate all unfilled spaces, enhancing the global surface quality. However, this improvement comes with the drawbacks of (i) hiding the core of these bundles for PLM and SEM observations and (ii) limiting the extraction of quantitative information using FM due to the local saturation with the fluorescence-enriched mounting epoxy resin, which is expected to cause localized brightness effects.

#### 2.2.3. Microstructure Characterization Using 2D Multimodal Imaging Techniques

The characterization of microstructural features in partially impregnated composite materials requires detailed information about the localization of the material constituents (i.e., GF and PP) and structural integrity correlated with manufacturing defects such as porosities, including trapped air bubbles or unsaturated zones termed dry zones. To this end, the current study focused on laboratory-scale 2D imaging equipment based on surface inspection using a comprehensive suite of imaging modalities. A numerical microscope (Axio Zoom V16, Zeiss^®^, Germany) equipped with a white light source, a fluorescence light illuminator (HXP 200 C), and a 20-megapixel microscope camera (Invenio20EIII, DeltaPix^®^, Denmark) was used. This setup is suitable for collecting RGB-encoded images from both polarized light microscopy (PLM) and fluorescence microscopy (FM) modes. This microscope, with an automated displacement-controlled stage and an X80 magnification lens, allowed for extended field imaging across all six layers of fiber bundle cross-sections throughout the sample’s thickness (Z-direction), covering between five and six complete bundle cross-sections within each layer. Precautions were taken to localize the bundle cross-sections at least by excluding one bundle from each of the cut edges (by the diamond saw) in the through thickness direction of the prepared composite sample. A scanning electron microscope (JCM6000, Jeol^®^, Japan) operating in Backscattered Electron Detector—Composition mode (BED-C mode) was used at X200 magnification to collect extended-field images (see Appendix A). SEM samples required additional gold-based metallization. The increased magnification for SEM observations was deliberate to maximize details at the scale of single fibers. The lack of an automated stage in the SEM device necessitated operator-assisted manual translations for capturing overlapping images. The collected images followed a snake-like column-based trajectory, allowing later assembly (i.e., extended-field image reconstruction) using a numerical stitching operation to generate a complete overview of the targeted microstructure. All image acquisitions from the three microscopy techniques (PLM, FM, and SEM) were carried out on composite samples polished up to 3 µm according to the preparation protocol (refer to Section 2.2.2). The integrative use of multimodal imaging techniques provides a robust framework for characterizing the microstructure of partially impregnated composite samples beyond the classical use of optical microscopy (OM) in Figure 4a. Indeed, PLM was used to control the state of the polished surfaces, augmenting contrast in anisotropic materials. PLM can reveal subtle differences on the observed surface, where birefringence indicates the orientation and integrity of fibers and the polished surface. This imaging technique excels in identifying areas with structural anomalies, such as surface scratches or rough zones, which are critical for assessing the quality of the polishing process (Figure 4b). FM is expected to enhance porosity visualization in composites by using a resin mount enriched with fluorescent dye. This resin infiltrates open porosities and/or dry zones. UV light excites the dye’s UV-sensitive molecules, which emit light at varying wavelengths based on the dye concentration (Figure 4c). The intensity of fluorescent-rich zones depends on the EpoDye concentration, corresponding to the depth of the damage, mainly due to process-induced porosities or, secondarily, to surface preparation-induced damage. SEM in the BED-C mode was used for the detailed compositional analysis of the polished surface of the composite samples. This technique involves directing an electron beam onto the sample surface, where heavier elements (such as glass fiber) cause more electrons to scatter back. The BED-C mode captures these backscattered electrons to generate high-resolution images that provide contrast based on atomic number variations. This method is especially effective for mapping the distribution of different phases (i.e., resin, PP matrix, and glass fiber) within the samples (Figure 4d). SEM in the BED-C mode delivers precise compositional mapping, essential for analyzing glass fibers.

This multimodal approach allows for a comprehensive analysis of the same microstructure, with each technique providing unique insights that contribute to a detailed assessment about the material. The data are encoded in a 2D spatial array (i.e., numerical images), where each pixel represents specific physical properties. Based on a comparative qualitative inspection of the same zone of interest within the exposed surface of the microstructure using the OM, PLM, FM, and SEM techniques as illustrated in Figure 4, surface defects such as scratches are predominantly discernible with PLM. For instance, surface scratches are clearly detectable in the residual polypropylene layer marked by the triangle symbol in Figure 4. SEM images are more reliable for glass fiber localization and the clear identification of the fiber bundle contours, as seen in the representative bundles B1, B2, and B3 illustrated in Figure 4. SEM significantly contributes to detecting single fibers, where small red arrows in Figure 4d point to a single fiber that was torn out due to polishing from a 90°-oriented bundle. The same single fiber is barely detectable using OM but can also be detected using PLM. Since the single fiber is not impregnated with the fluorescent resin, it is undetectable using FM. FM clearly highlights the zones of localization of the fluorescent agent, as seen from bundle B3 in Figure 4, where high brightness branches indicate a significant presence of the fluorescence-enriched epoxy, indirectly indicating a poor impregnation of the fiber bundle core by PP. Bundle B2 in Figure 4 appears to have a high level of impregnation, as the presence of bright green color in Figure 4c is faint, with numerous black spots indicating exposed glass fiber surfaces. This observation is supported by corresponding OM and SEM observations.

## 3. Post-Processing Multimodal Images

The subsequent sections detail a workflow for post-processing multimodal images, with a primary emphasis on SEM and FM extended-field images, progressing to bundles of glass fibers. Figure 5 depicts this workflow at the macro-scale of the full thickness of the microscopy-inspected samples. It is worth noting that no filtering, histogram normalization, or brightness corrections were applied to the raw images (Appendix A) to preserve the authenticity of the captured data. The process begins by stitching single-tiled images from each imaging techniques to form extended-field images (cf. Section 3.1). Given that extended-field images based on PLM and FM have an X80 magnification, SEM images are resized to match this magnification by reducing their original X200 magnification, typically using interpolation methods to maintain image quality. All full-scale and extended-field images from the same composite sample are then registered to ensure their accurate alignment with the reconstructed extended-field SEM image as the reference image (cf. Section 3.2). These registered images are then compiled into a stack and subsequently cropped. The cropped FM and SEM images are subjected to pixel classification using Random Forest classifiers, a machine learning technique that classifies pixels based on various features extracted from the images (cf. Section 3.3). The processed images are stored in a five-layer stack.

At the meso-scale, this stack for each composite sample is manually annotated to identify and segregate fiber bundles oriented at 0° into distinct five-layer stacks. Manual annotation follows strict criteria to ensure consistency, with each stack meticulously inspected for stitching imperfections and undergoing local adjustments to ensure precise alignment. The final bundle images are compiled into a dataset containing about 15 individual 0°-oriented bundles from each composite sample. This quantity is expected to create a robust dataset for both qualitative and quantitative analyses. Additional details on the operations within this two-scale workflow, including the specific algorithms and software used for stitching, registration, and pixel classification, will be elaborated upon in the following sections.

### 3.1. Macro-Scale Stitching: Reconstruction of Extended-Field and Full-Scale Images

Stitching operations enable the merging of elementary images to visualize larger areas of interest while preserving the pixel resolution of the input elementary (i.e., tiled images). This method is essential for conducting principally qualitative analyses at the macro- and meso-scales (i.e., physical scale of the plate thickness and the respective individual glass fiber bundles) and then achieving the precise pixel classification of pixels and segmentation of microstructures principally at the micro-scale of the single glass fibers. An illustrative example of the considered image stitching method is provided in Figure 6, which depicts a stitched grid of (3 × 3) SEM local images, including the acquisition path of these images. During the acquisition, each image included an overlapping area ranging between 10 and 25%, covering identical microstructural details, such as single fibers, to aid in creating the extended-field image. This stitching method utilizes the open-source Image Stitching of ImageJ/Fiji plugin [28,29], which relies on the Fourier Shift theorem to compute in-plane (X,Y) translations between a grid of 2D images. The corresponding computations are based on a predetermined path for collecting these images and leverages cross-correlation measurements to determine the optimal overlap, thereby reconstructing extended-field images from numerous tiled input images. The Image Stitching plugin requires prior inputs, including the number of columns and rows, the expected overlap range between tiles, and the tolerated minimum and maximum displacement thresholds. The plugin can automatically compute the optimal overlaps, and then merges the images based on different methods, including the linear blending of gray levels within the common intersection areas between adjacent images. It is this same procedure that was applied for generating the full-field stitched images presented in Figure 7 based on the corresponding elementary tiled images provided in Appendix A.

### 3.2. Macro-Scale: Resizing and Registration Operations of Extended-Field Images

Utilizing the same Zeiss microscope for both PLM and FM simplified the registration process due to the similarity in acquiring the grids of tile images (Appendix A), thereby facilitating the alignment of their correspondent extended-field images. SEM images were resized for congruence with the sample’s plate thickness, as determined by the PLM reference image. The resized SEM images, based on bicubic interpolation using ImageJ/Fiji functionality, offered enhanced clarity and contrast for visualizing glass fibers and served as reference images for the rigid registration of the corresponding PLM and FM images. This registration involved translations in the XY plane and rotations about the out-of-plane axis. The registration process aligned control points, primarily individual glass fibers and the outer shapes of fiber bundles. Rigid registration operations utilized the TrakEM2 plugin from ImageJ/Fiji, which supports both the manual and semi-automatic alignment of image stacks through rigid transformations. These transformations can be stored as “.xml” files, and the aligned images can be exported post-registration to the same input image-encoding formats. Additional details about the TrakEM2 plugin are available in [30]. Post-registration, the images were grouped into stacks of images (i.e., a multilayer image) and then cropped to eliminate extraneous zero-value pixels at the edges and to focus on a region of interest at the macro-scale containing between five and six complete fiber bundles in the top layer of the composite plate (referred to as layer 1 and containing 0°-oriented bundles). Figure 7 provides a visual summary of the resized and registered images for the three analyzed samples, while Appendix A outlines the dimensions and pixel sizes of the images post-stitching, resizing, registration, and cropping.

The obtained extended field PLM images, as illustrated in Figure 7a–c, facilitate the qualitative inspection of the polished material surfaces. These images reveal the presence of numerous scratches, particularly within polypropylene-rich areas and fluorescence-enriched epoxy resin layers used for embedding the composite sample. Based on PLM images and SEM scans of the 3 µm polished surface according to the four-step surface preparation procedure (see Section 2.2.2), the presence of single fibers pulled out from the 90°-oriented bundles and covering the adjacent 0°-oriented layers of fiber bundles is absent. The absence of such debonded single fibers on the final surface post-polishing qualitatively indicates the improved impregnation of dry zones within the exposed composite layer by polishing, thus underscoring the efficacy of the developed four-step polishing technique. Additionally, a localized examination of the fiber bundles showed an acceptable level of circularity for individual fibers within the 0°-oriented bundles (check zones A, B, and C in Figure 7).

### 3.3. Macro-Scale: Random Forest-Based Pixel Classification

The pixel classification methodology was applied using the open-source software Ilastik (version 1.3.3) [31,32], developed by Sommer et al. [33], which incorporates integrated machine learning approaches designed for users with limited machine learning experience. This method employed Random Forest classifiers for analyzing high-dimensional image data [34]. Ilastik accommodates both mono-channel (8-bit) and multichannel (RGB) inputs. Initially, users select up to nine data features, including color/intensity, edge detection, and texture, which Ilastik automatically computes at various scales using Gaussian smoothing variance. Users then graphically select and label a few pixels using a virtual brush tool, assigning specific labels to each pixel class. These annotated pixels and their associated features are used by the Random Forest classifier to generate a decision tree matching the training labels, allowing for real-time feedback and iterative improvements to enhance the classifier accuracy. Ilastik outputs label predictions and accuracy levels, assisting in the evaluation of classifier performance. To optimize classifier convergence, random pixels from each extended field image were selected to adjust for variations in contrast and brightness, as detailed in Figure 7. Computations were executed on a Windows workstation with an Intel Xeon E5-1650 v2 CPU, 64 GB RAM, and an Nvidia Quadro K2000 GPU, completing the classification process within approximately 30 min per extended-field image. In the case of SEM extended-field images, five specific pixel labels were utilized: center of single glass fibers, edge of glass fibers, zones of rich PP such as residual PP layers, fluorescence-enriched epoxy outside the sample, and unfilled spaces at the ground level of 8-bit grayscale images (see Figure 7). Given that the study’s primary focus is quantitative analysis, pixel classification in SEM images primarily targeted pixels at the center of glass fibers to facilitate separation within densely packed 0°-oriented fiber bundles, enabling the calculation of local distances to neighboring fibers and the generation of distance maps between the centers of single glass fibers. Figure 8 shows input SEM images and segmented glass fibers, noting local contrast variations due to the sensitivity of the retro-diffused electron probe during image acquisition. Despite these variations, the high gray level values of glass fibers in 8-bit SEM images enabled accurate pixel segmentation (see Figure 8, zones A, B, and C). These contrast variations were considered non-critical for the Random Forest classifier-based machine learning pixel classification. Post-segmentation, SEM images were used to delineate the physical limits of fiber bundles and systematize the nomenclature from left to right for complete bundles and from top to bottom between the layers of UD plies principally oriented perpendicularly to the observation plane (Figure 8a–c). The similarity in nomenclature designation (L_i_M_j_) of bundles does not imply any bundle order correspondence between the composite samples; it is used solely for simplicity.

In RGB-encoded FM extended-field images, pixel classification utilizes a five-label scheme through Ilastik with a Random Forest classifier, focusing on the intensity of the green color to indicate fluorescent agent concentrations. This methodology categorizes fluorescence into five levels: 0 (no agent), 1 (low saturation), 2 (medium-low saturation), 3 (medium-high saturation), and 4 (maximum brightness indicating pixel saturation). Level 0, depicting the absence of fluorescence, is prominently seen in Figure 9a, highlighting the sample Cr_0% known for high-quality impregnation and minimal dry zones. Level 4, often at the interface between the robing resin and the composite samples as shown in Figure 9a–c, represents maximum brightness and potential optical interference effects. Level 1 indicates partial staining on PP layers, likely due to abrasive polishing, signifying thin residual layers of the embedding matrix or slightly stained PP. Level 2, associated with medium-low saturation, typically appears in resin zones outside the main composite structure. Level 3, characterized by a yellow-dominated hue representing medium-high saturation, is visible in the 90°-oriented layers in both Figure 9b,c, marking dry zones or debonded and broken single glass fibers, as identified by the scratching marks on the PP layers depicted in Figure 7a. Figure 9 provides a comprehensive view of these fluorescence levels across three composite samples, with segmented images in Figure 9d–f illustrating variations in impregnation levels and resin distribution. This classification framework aids in the precise segmentation and analysis of the composite microstructure. Due to limited visibility in 90°-oriented fiber bundles caused by excessive fluorescent resin residues from polishing, qualitative inspections will focus on 0°-oriented bundles.

In Figure 9d, the Cr_0% sample shows predominant levels 0 and 1, indicating limited fluorescent agent presence and suggesting well-exposed single glass fibers with minimal polishing damage and high saturation by PP from the film-stacking manufacturing method. In contrast, the Cr_30% sample in Figure 9b presents high impregnation quality in 0°-oriented layer 5, adjacent to the bottom PP layer, likely due to PP fluidity during manufacturing. Progressing from bottom to top through layers 5 to 3 and then to 1, mixed levels of 1, 2, and 3 indicate partial impregnation. As the compaction ratio increases from Figure 9e–f, the impregnation quality appears to improve, although level 1 bundles near the top exhibit higher impregnation at the edges, suggesting limited through thickness polymer flow. This analysis potentially supports a justification of the through thickness flow in a simplified-CRTM process, though further analysis is required to correlate macro-scale compaction ratios and impregnation scenarios in the samples.

### 3.4. Meso-Scale: Workflow Applied to 0°-Oriented Bundles

The main focus of the current section based on meso-scale inspections is focused on 0°-oriented fiber bundles to conduct the quantification of glass fiber content, porosity, and consequently, polypropylene as a third possible component. In the specific context of conducting meso-scale quantitative analyses of the segmented images obtained from FM and SEM, all image processing operations relied on finalized extended-field image and their respective segmented images illustrated in Figure 7, Figure 8 and Figure 9. All post-processing procedures conducted at the meso-scale of fiber bundles were based on open source Fiji/ImageJ software, open source Ilastik (version 1.3.3), and MATLAB R2022a software (Mathworks Inc., MA, USA). 

### 3.5. Meso-Scale: Inspection of Stitched Images

First, the manual annotation and extraction of fiber bundle contours were conducted based on local inspection at the meso-scale of the extended field output images following the macro-scale image post-processing workflow, including stitching, resizing, rigid registrations, cropping, and pixel classification operations. The main challenges were related to: (i) controlling the output of automatic stitching operations of the large grids of collected tiled images (see Appendix A) and (ii) checking the local precision of the rigid registration operations conducted at the macro-scale. Indeed, the considered macro-scale control procedure was based on the extended field images from the three microscopy techniques (PLM, FM, and SEM), as illustrated in Figure 7. First, the three images of the same microstructure generated from the three microscopy techniques (PLM, FM, and SEM) and the segmented images of SEM and FM were all merged into a five-layer single stack (i.e., pile of 2D images), all encoded into the RGB format. Then, manual annotations were conducted using a polygon-based manual contouring of fiber bundles using ImageJ/Fiji to extract all individual bundles from the images based on the visual delimitation of each single bundle limit based on SEM images.

The visual inspection of each bundle was then conducted to check for the presence of critical stitching imprecisions. Subsequently, rigid registration was applied to the five-layer stack (of SEM, PLM, FM, SEM segmented, and FM segmented images) to further correct any potential imprecision at the scale of the single fibers. The main critical imprecision consisted of misalignment at the interface of overlapping individual tiles, causing a shadow effect or the local discontinuity of the microstructure. The detection of such imperfections is considered an exclusion criterion for a few fiber bundles, which will not be included in the following quantitative analyses. In this study, only the bundles L1M6 and L5M5 extracted from the sample Cr_0% were excluded, and the corresponding imperfections are illustrated in Figure 10. This problem of limited precision in stitching is expected when using large datasets of individual tiles. The rest of the bundles are considered within the required precision range to conduct the quantitative analyses. In the context of FM and PLM images, the image acquisition included the automatic coordinate-based collection of individual tiles, and it was noted that stitching misalignments were absent with respect to the considered stitching approach using the Image Stitching plugin of ImageJ/Fiji. A second challenge observed after macro-scale stitching is the presence of local changes in grayscale brightness levels between the tiles, particularly in SEM images, as seen in the macro-scale extended-field images in Figure 7. This effect is due to long-duration acquisitions ranging from 8 to 16 h (according the total grid for each sample indicated in Appendix A), which are associated with inevitable drift of the incident electron beam and performance stability of the electron detector of the used SEM equipment. As indicated earlier, no histogram normalization or filtering operations were applied to the raw tiled images or the reconstructed extended-field images. To check the performance of Random Forest classifiers in overcoming such effects, a representative bundle, L5M3, from the Cr_0% sample was considered. Detailed visuals of the segmentation procedure are provided in Figure 11. Two regions of interest, ROI 1 and ROI 2 (Figure 11a), were extracted from the raw SEM image of the bundle, and the corresponding histograms of grayscale levels were generated (Figure 11b). The obtained histograms show that equipment drift over time (as the right and left sides of this particular bundle are from two different tiled images) causes a shift in histogram peaks from left to right, with the peak around grayscale level 250 corresponding to glass fiber pixels and the peak around grayscale level 50 corresponding to polymer-rich zones. This shift is considered insignificant during pixel classification, as shown by the classified pixels from ROI 1 and ROI 2 (Figure 11c), and mainly affects the distinction between PP and the epoxy-based mount resin. This intensity-related local variability was deemed negligible in this study. However, as a precaution, quantitative analyses will evaluate the glass fiber content from each bundle to estimate the uncertainty caused by such brightness change-related image artifacts.

After conducting all the inspection operations, a clear visual overview of all extracted individual bundles oriented at 0° from SEM and FM extended-field images is presented in Appendix A. The PLM images were mainly collected to check the surface state following the polishing procedure and will not be used during the meso-scale quantitative analyses, and the corresponding single bundles are not illustrated.

### 3.6. Meso-Scale: Quantitative Analysis Workflow of 0°-Oriented Fiber Bundles

The conducted quantitative analyses were based on the extracted bundle SEM, FM, and corresponding segmented images. First, pixels corresponding to the central pixels of single glass fibers were extracted and converted to binary masks, as illustrated in Figure 12c. In a second step, the SEM images of fiber bundles were subjected to a two-step procedure for the object classification of the extracted single fibers using the open-source Ilastik software (Version 1.3.3), which allows the attribution of a single identifier to each fiber (Figure 12d). The objective of this operation was to check the number of single glass fibers within a bundle and to extract the center of each fiber. Based on interpolation between the coordinates of the fiber centers, an initial bundle contour was identified. This contour overestimated the bundle area, particularly in bundles with non-packed fibers, such as those from the top layer of the composite sample Cr_41%, where the limited impregnation of layer 1 (see Figure 7) was discussed in the section on the qualitative inspection of the corresponding macro-scale-segmented FM image. To overcome this limitation, normalized distances between the center of each single glass fiber and its nine adjacent neighboring fibers were evaluated using MATLAB, and the corresponding maps of normalized distances were provided, as illustrated in Figure 12e and in Appendix A. Maps of normalized distances were first considered without any filtering to define the largest bundle contour passing by the center of all circumferential single fibers. In a second step, a maximum normalized distance threshold of 0.4 was judged appropriate by the authors to objectively define a narrower contour for all fiber bundles, excluding parts of single fibers with normalized distances higher than 0.4, while conserving the shape of the packed bundles. An illustration of the two generated contours, one large and one narrow, is shown in Figure 12f and in Appendix A. These two contours were then superimposed on the segmented FM image of the bundle to extract the pixels within each contour representing levels 2, 3, and 4 of fluorescent concentration. These levels were considered indicators of limited impregnation, based on the macro-scale qualitative analysis of the segmented images. Next, the pixels corresponding to the segmented glass fibers in Figure 12c were subtracted from the masks representing levels 2, 3, and 4 of the FM images to define the zones outside the single glass fibers, representing either porosity or areas of limited impregnation. A second subtraction of the total bundle area, defined by each of the previously identified contours in Figure 12f, removed the glass fiber pixels and the limited impregnation pixels, leaving the remaining pixels to represent the PP-impregnated areas. This meso-scale workflow resulted in two sets of quantitative values for each bundle (one from the large contour and one from the narrow contour). The same process was repeated and generalized to all bundles identified as suitable for quantitative analysis. In this context, all distance maps and the two contours are provided in Appendix A.

## 4. Results

### 4.1. Quantification of GF Single Filaments

Following the pixel classifications of SEM images, an automated object classification procedure was applied to the population of pixels representing the central core of single fibers. An additional constraint was imposed on the size of isolated pixel clusters to exclude fragmented cross sections, separate adjacent cross sections of single fibers, and identify non-separated single fibers exceeding an area threshold of 650 pixels. The conducted flow chart was based on Ilastik and did not require the implementation of any Appendix A. Based on the separated objects, we extracted the coordinates of the center of each single glass fiber and quantified the number of single fibers per bundle. A summary of the collected data is provided in Figure 13. The global mean (μ) of single filaments based on all fiber bundles was 4038, with a standard deviation (σ) of 63. Most of the counted filaments fell within the 95% confidence interval, corresponding to (μ±2×σ), with approximately 5% of data points falling outside this range. The filaments falling outside the (μ+2×σ) interval were due to the presence of stitching fibers imbricated in the 0°-oriented bundle, such as in bundle L1M4 from plate Cr_41%, or the significant presence of fragments from glass fibers imbricated between the circular cross-sections of the 0° bundle filaments, such as in bundles L1M3, L2M3, and L3M3 from the same composite plate. These observations highlight the limits of the defined surface preparation procedure, where mechanical polishing cannot completely eliminate the breakage of GF tips. The maximum relative error computed from these identified bundles with a high number of single filaments was 3.87%. Conversely, a few fiber bundles presented a low number of single filaments, falling below the lower bound of (μ−2×σ), such as bundle L5M3 from plate CR_0%, which had 3900 single filaments, representing a relative error of 3.42%. Beyond experimental bias that may explain variations in the number of counted single filaments, this operation was also conducted to estimate the precision of the object identification procedure in identifying single filaments. The obtained data appear to fall within the 95% confidence interval.

### 4.2. GF Area Fraction Quantification

Glass fiber (GF) area fractions within the bundles oriented at 0° were quantified to assess the sensitivity of the SEM image segmentation to local brightness variations in the BED-C mode. The obtained quantitative data are graphically represented in Figure 14. For the sample produced by film stacking, the average GF area percentages in layers 1, 3, and 5 were 46.6% (±3), 46.5% (±2.8), and 44.8% (±2.2), respectively. The percentages extracted from the narrow contours were 47.4% (±3.0), 47.1% (±3.0), and 45.6% (±2.0). As the fiber bundles in this configuration are expected to have a high level of impregnation, the quantified surface areas are consistent. For the plate Cr_30%, the average GF area percentages in layers 1, 3, and 5 were 53.7% (±0.6), 57.4% (±0.4), and 57.3% (±0.4), respectively. The percentages extracted from the narrow contours were 54.6% (±0.5), 57.9% (±0.6), and 57.7% (±0.4). For the plate Cr_41%, the average GF area percentages in layers 1, 3, and 5 were 51.1% (±1.7), 57.7% (±1.1), and 57.8% (±1.1), respectively. The percentages extracted from the narrow contours were 52.7% (±1.7), 58.3% (±1.1), and 57.7% (±1.2). For the samples extracted from the plates manufactured using simplified-CRTM, the average GF area percentages were within the same ranges for both samples. However, the bundles from layer 1 for both samples contained less GF than layers 3 and 5.

This observation can be attributed to a higher exclusion of loose GF, reducing the bundle contour and the number of single GFs included. The global observations did not show significant variations due to contrast differences in the SEM images used to reconstruct the extended field of view. This indicates that the Random Forest classifiers are less sensitive to these image biases. The variation in percentage between Cr_0% and the other plates can be explained by the morphology of the extracted bundles, which are less flattened in Cr_0% compared to the other plates.

### 4.3. Quantification of Porosity Area Fractions Based on Narrow and Large Bundle Contours

Porosity area fractions within the 0°-oriented bundles were quantified based on the large and narrow bundle contours and are reported in Figure 14. For the sample produced by film stacking, the average porosity area percentages in layers 1, 3, and 5 were 5.2% (±1.8), 0.02% (±0.04), and 3.8% (±3.4), respectively. The percentages extracted from the narrow contours were 4.0% (±1.7), 0.02% (±0.03), and 3.6% (±3.2). These results indicate minimal porosity within the core of the composite plate, suggesting a high degree of impregnation. For the plate Cr_30%, the average porosity area percentages in layers 1, 3, and 5 were 34.9% (±2.8), 35.5% (±3.5), and 12.4% (±4.24), respectively. The percentages extracted from the narrow contours were 33.7% (±2.6), 34.9% (±3.5), and 11.9% (±4.4). The average porosity levels in layers 1 and 3 of plate Cr_30% were similar, whereas layer 5, which was closest to the PP-rich bottom zone, had about 11.9% porosity. The bundles in layer 5 showed significant impregnation but were not fully saturated. For the plate Cr_41%, the average porosity area percentages in layers 1, 3, and 5 were 24.1% (±5.7), 13.6% (±5.5), and 0.2% (±0.1), respectively. The percentages extracted from the narrow contours were 21.5% (±6.3), 12.0% (±5.3), and 0.1% (±0.1). Plate Cr_41% displayed a clear trend of a through thickness impregnation gradient from the bottom to the top layers compared to plate Cr_30%. Layer 5 had a negligible amount of porosity, while a significant increase was observed from layer 5 to layer 3 to layer 1. This observation was confirmed by the qualitative inspection of the extended field images, where the bundles of plate Cr_41% showed loose fibers towards the outer surface, indicating the presence of PP dry zones.

### 4.4. Statistical Significance of the Generated Data: One-Way Anova Test

The purpose of this section is to assess the statistical significance of the obtained data in detecting the effect of manufacturing configurations on the variation of the degree of impregnation of 0°-oriented bundles between layers 1, 3, and 5. For this purpose, a one-way analysis of variance (ANOVA) at a 95% confidence interval was conducted. The obtained data, illustrated in Figure 14, were converted into the degree of impregnation of a bundle, defined by Equation (2), which allows the estimation the saturation of the unfilled spaces within the fiber bundle cross-section with the PP matrix after the exclusion of the area filled by single GF filaments.
(2)$$Degree\;of\;impregnation=100\times \frac{Total\;Bundle\;area-GF\;area-Porosity\;area}{Total\;Bundle\;area-GF\;area}$$

#### 4.4.1. Hypothesis 1: Consideration of All Fiber Bundles without Any Distinction between Composite Layers

The null hypothesis for ANOVA considers that no differences among the mean values of impregnation degrees between the layers from plates Cr_0%, Cr_31%, and Cr_40%. The decision rule based on ANOVA considers the rejection of the null hypothesis if the p-value is lower than or equal to the significance level. The significance level is commonly defined at 0.05 as set in another study by the authors (i.e., meaning that there is a 5% risk of concluding that there is an effect) [35]. Since the current study is exploratory, a less stringent level of risk defined at 10% was considered to increase the test’s sensitivity. Table 3 represents the average degree of impregnation of bundles layer per layer as well as the manufacturing configurations. The obtained p-values from the one-way ANOVA were evaluated based on data evaluated from the large and narrow contours. The computed “*p*-values” that were obtained were, respectively, 0.053 and 0.0463 for the large and narrow contours. With consideration of a risk level of 5%, the p-value for the narrow contour of fiber bundles successfully rejects the null hypothesis. For the case of the large bundle contour, the p-value, despite being close to the 5% limit, technically fails to reject the null hypothesis, indicating no conclusive significance of the manufacturing conditions on the impregnation of fiber bundles from layers 1, 3, and 5. However, with the increase in the sensitivity of the test, as due to the limited number of bundles per layer and due to the variability during the manufacturing of composite materials, the null hypothesis can be rejected, meaning that the three manufacturing methods have different impregnation effects on the bundles from layers 1, 3, and 5. This less restricted test limit of 10% can be supported by the qualitative observations from the bundles, but, on the other hand, it indicates that there is a need for the testing of the image-based approach with samples manufactured with a higher level of confidence of control of the manufacturing process, which is an ongoing target by the authors.

#### 4.4.2. Hypothesis 2: Considering the Distinction between Composite Layers

With consideration of a separation between manufacturing conditions and the layers from which bundles are extracted, the null hypothesis corresponding to the ANOVA analysis performed separately for each layer posits that there is no significant difference in the means of the porosity values across the different manufacturing methods (A, B, and C) for each layer. Specifically, for each layer, the null hypothesis asserts that the average porosity is the same regardless of the manufacturing method used: For layer 1, the null hypothesis (H0) states that the mean porosity for the manufacturing conditions of plate Cr_0% (Method A) equals the mean porosity for the manufacturing conditions of plate Cr_30% (Method B) and the manufacturing conditions of plate Cr_41% (Method C); similarly, for layer 2 and layer 3, H0 maintains that the mean porosity for Method A equals that of Methods B and C. Conversely, the alternative hypothesis for each layer suggests that at least one of the manufacturing methods has a different mean porosity compared to the others, indicating a significant effect of the manufacturing method on porosity within each layer. As indicated from results from the ANOVA analysis in Table 3, all *p*-values in this context are equal to zero, meaning a strong rejection of the null hypothesis and confirming a high variability index on the porosity levels of each layer while changing the process.

## 5. Assessing Uncertainty in Porosity Quantifications from FM- and SEM-Based Approaches

To further consolidate the previous results, an evaluation of the degree of uncertainty was conducted based on a quantitative assessment of the surface area percentage (SAP) of dry zones (including porosity and unfilled space within the preform) obtained from the multimodal approach, including (i) FM images from Figure 9, concentration levels 2, 3, and 4 of the fluorescent agent and (ii) SEM images from Figure 8, pixels corresponding to GF. In the case of SEM only-based data, only pixels corresponding to unfilled zones and epoxy mount, as indicated in the segmentation in Figure 8, were used. Continuing the previous sections, the analysis was limited to the 0°-oriented cross-section of the fiber bundles. The data obtained were added to Figure 14, with the legend modified to include additional surface area percentages of porosities based on SEM images without affecting the data. For the Cr_0% sample, the SEM-based quantification of the surface area percentage (SAP) of porosity (dry zones), as shown in Figure 14a, is higher than dry zones infiltrated by the fluorescent agent-enriched mount epoxy, according to the multimodal-based quantification method. Specifically, the average SAP of porosity based on SEM is approximately 23.4% (±2.3%), while the multimodal-based measurement relying on FM is about 2.9% (±3.0%). When using a tighter contour of the meshes (in red color in Figure 12f), the average values for both cases are not significantly different. As observed from Figure 9a and its corresponding segmented FM image in Figure 9d, there is no significant presence of high concentrations (levels 2 to 4, from Figure 9d) of the fluorescent agent in layers 1 and especially layer 3 of the bundle cross-sections oriented at 0°. This qualitative verification suggests that the SEM-based quantification may overestimate the surface area percentage of porosity (dry zones) due to the limited grayscale distinction between PP and epoxy mount using the BED-C mode, as indicated by the histogram peaks in Appendix A (for grayscale levels ranging between 25 and 150). Indeed, the signal drift during SEM acquisitions, while collecting the 850 tiles in the case of the Cr_0% sample, is associated with local changes in the gray levels of PP and mounting epoxy between the tiles, making it challenging for user-assisted Random Forest classifiers to accurately distinguish between PP and epoxy-rich pixels (check segmentation details B and C in Figure 8d). On the other hand, as the Cr_0% sample was extracted from a plate manufactured based on an alternated film stacking according to configuration 1 (Figure 1c), it is expected that PP impregnates easily the adjacent GF plies. With consideration of all previous observations related to the Cr_0% sample, the uncertainty level is higher based on the SEM images compared to the multimodal approach integrating FM-based quantification of the surface area percentage of porosity based on the trace of infiltrated fluorescent agent under UV light.

For the 0°-oriented bundles in the plate Cr_30%, the SEM-based surface area percentage (SAP) of porosity (dry zones) is approximately of 6.1% (±2.6%). The SAP of porosity (dry zones) based on the FM quantification approach is around 28.1% (±11.4%), using the green contour of the GF bundles. In addition, the histogram of grayscale levels for the macro-scale SEM image of the Cr_30% sample in Appendix A shows at least four distinctive peaks for gray levels ranging from 25 to nearly 125. These peaks indicate a significant gray level drift during the acquisition of the corresponding 348 tiles, making it challenging to accurately differentiate between PP and epoxy, similar to the Cr_0% case. In contrast, the FM image in Figure 9b and its corresponding segmented image in Figure 9e show high concentrations of the fluorescent agent. The FM-based quantification approach of porosity also demonstrates sensitivity to the degree of impregnation, as seen in the SAP of porosity (dry zones) in layer 5, which is closest to the PP layer in the plate manufactured according to configuration 2 (Figure 1c), compared to layers 1 and 3, which are expected to be less impregnated as they are more distant than layer 5 from the pool of melt PP during manufacturing.

The SEM-based quantification of SAP of dry zones seems to underestimate the porosity level in the 0°-oriented bundles. The Random Forest classifier appears to overestimate the pixels corresponding to PP compared to those corresponding to the epoxy used as a resin mount, especially in layers 1 and 3, with no significant change in SAP between bundles extracted from layers 3 and 5, as shown in Figure 14b. For bundles corresponding to layer 5 from the Cr_30% sample, the average SAP of porosity from SEM-based quantification is about 3.9% (±0.9%), while the SAP based on the FM approach is approximately of 12.4% (±4.24%). This suggests that, for 0°-oriented bundles with a relatively high impregnation level, notably layer 5, the FM quantification approach provides roughly a one order of magnitude greater accuracy compared to solely using SEM. For 0°-oriented bundles with limited impregnation levels, such as in layers 3 and 1, the risk of underestimation of the SAP of porosity when relying only on SEM images is around 20%. This estimation considers the mean SAP for layers 1 and 3 in the Cr_30% case, which are of 7.1% and 35.2%, respectively. When compared to relatively high impregnation bundles extracted from the Cr_0% sample, SEM seems to have about 807% overestimation of porosity levels compared to the FM-based approach. These estimations highlight the sensitivity of SEM acquisitions and the difficulties related to pixel classifications, especially for PP and epoxy based on the BED-C mode. This issue requires further investigation into correcting the signal brightness of collected SEM tiles and quantifying the error propagation due to acquisition artefacts to assess the uncertainty level of segmenting PP and epoxy in the context of epoxy-based resin mounts frequently used in microstructure characterizations of thermoplastic composite materials. These suggested investigations were considered to be outside the scope of the current study.

In the case of fiber bundles in sample Cr_41%, Figure 14c indicates less disparity in the SAP of porosity (dry zones) compared to samples Cr_0% and Cr_30%. These observations align with the assumptions made about layer 5 of the Cr_30% sample. Specifically, for the Cr_41% sample, the SAP of porosity based on SEM images for layers 5, 3, and 1 were 5.3% (±2.5%), 5.2% (±1.5%), and 12.9% (±1.3%), respectively, compared to 0.2% (±0.13%), 13.6% (±5.5%), and 24.1% (±5.7%) based on the FM-based approach. These values indicate an underestimation of porosity levels of 38.2% for layer 3 and 53.5% for layer 1 based only on the SEM image. The uncertainty levels for layers 1 and 3 seem consistent with layer 5 of the Cr_30% sample. However, for layer 5 of the Cr_41% sample, there is an overestimation level of 2650% of SEM-based SAP of porosity compared to the FM-based approach. This observation is in the context of the high impregnation level of layer 5 for the sample Cr_41%; thus, FM can also be considered to underestimate the SAP of porosity (dry zones) due to the limited infiltration of the fluorescent agent-enriched epoxy mount and the consideration of fluorescence concentration levels higher that level 2, as indicated by the scale of concentration levels provided in Figure 9. This overestimation of porosity by SEM compared to the FM-based approach aligns with the Cr_0% sample, where most layers are expected to be well impregnated. The case of layer 5 in the Cr_41% sample indicates that the multimodal approach associating FM and SEM techniques can be reliable for quantifying the SAP of porosity in partially impregnated bundles; however, more precautions are required, especially in the case of fully impregnated bundles.

## 6. Conclusions

This study highlighted the potential of combining multimodal and extended-field imaging, principally integrating PLM, SEM, and FM techniques, for inspecting the surface quality after mechanical polishing and for assessing the degree of impregnation at the meso-scale of 0°-oriented bundles of glass fibers. The multimodal approach focused on FM and SEM extended-field images, providing a robust characterization of fiber bundles based on SEM images quantifying single fibers and area fractions of GF, and porosity based on the FM extended-field analysis of the concentration levels of the fluorescence-enriched epoxy resin mount. The methodology involved detailed experimental workflows for surface preparation to minimize defects during the mechanical polishing of partially impregnated polymer matrix composites. Two-scale post-processing workflows of multimodal images were developed, including image alignment and pixel classification techniques at the macro-scale, ensuring precise quantitative analyses at the meso-scale of fiber bundles oriented perpendicularly to the polishing plane (0°-oriented bundles). Bundle contours at the meso-scale were objectively identified based on normalized distance maps following the object classification of single glass fibers. Large and narrow contours were defined, with the large contour encompassing all single glass fibers and the narrow contour including only single fibers with a normalized distance to their nearest neighbors lower than 0.4 to exclude isolated single fibers detached from the bundles. The image post-processing workflow at the meso-scale was comprehensively evaluated to assess the number of single GF filaments per bundle, the area fractions of glass fibers (GFs), and porosity in 0°-oriented fiber bundles across three manufacturing configurations. Analyses utilizing pixel classifications of SEM images, followed by an automated object classification procedure, revealed that full-scale extended-field images at the considered resolutions indicated an average of 4038 single filaments per bundle, with a standard deviation of 63, indicating that most data points fell within the 95% confidence interval. The GF area fractions indicated consistent impregnation levels for the film stacking sample, with percentages ranging between 44.8% and 47.4%, indicating fewer GF fractures. Higher GF area fractions were observed in samples Cr_30% and Cr_41%, suggesting less significant impregnation levels by the simplified-CRTM manufacturing process with changing compaction ratios, but the levels were, respectively, around 53.7% and 57.3%, indicating relatively controlled single fiber damage following the four-step surface preparation protocol. Porosity area fractions were minimal in the film stacking sample, whereas higher levels were detected in Cr_30% and Cr_41%, with distinctive impregnation gradients noticed between layers 1, 3, and 5 in Cr_41%, reflecting more representative through thickness flow impregnation. The statistical analysis using a one-way ANOVA confirmed significant differences in impregnation degrees across different manufacturing configurations, particularly at a 10% risk level. Furthermore, the separate layer analysis demonstrated significant variability in porosity levels due to manufacturing conditions, with all p-values being zero. Overall, the findings underscore the significant impact of multimodal imaging for the quantitative analysis of the degree of impregnation and porosity within fiber bundles. Regarding manufacturing configurations, the statistical results from the current study and the datasets of fiber bundles indicate the need for more precise control in the manufacturing process to achieve more consistent process-related correlations. Additionally, the efficacy of multimodal imaging as a powerful tool for detailed inspection and quality control in composite material production was demonstrated. Its implementation could lead to a better understanding and optimization of compression molding and CRTM manufacturing processes, which constitute an active research area for the authors. The current study also generated a significant dataset of multimodal images of 0°-oriented fiber bundles, which opens the door to interdisciplinary topics related to machine learning-based image post-processing approaches in the context of thermoplastic matrix composites.

## Figures and Tables

**Figure 1 polymers-16-02171-f001:**
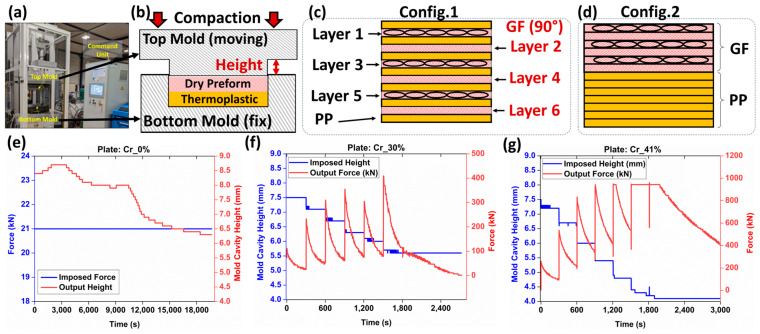
(**a**) Manufacturing platform: Pinette PEI press. (**b**) Schematic representation of the mold used. (**c**) Film stacking configuration 1 used for compression molding. (**d**) Film stacking configuration 2 used for simplified-CRTM. Recorded force-mold cavity heights for plates: (**e**) compaction ratio (Cr), Cr_0%, (**f**) Cr_30%, and (**g**) Cr_41%.

**Figure 2 polymers-16-02171-f002:**
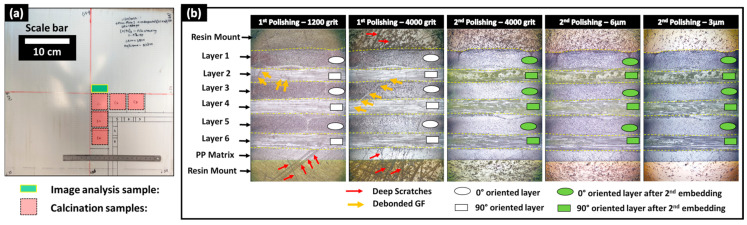
(**a**) Locations and geometrical sizes of extracted microscopy and burn-off samples. (**b**) Evolution of surface state during polishing operations: after the first (white symbols) and second (green symbols) applications of fluorescent dye-enriched resin mount.

**Figure 3 polymers-16-02171-f003:**
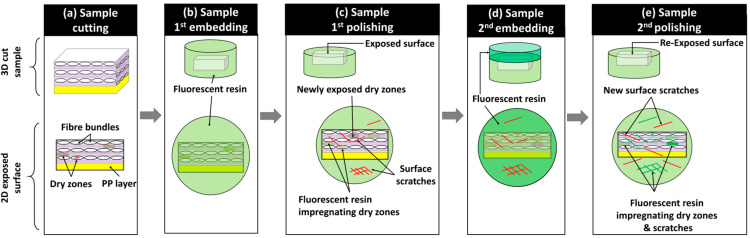
Synthetic overview of the proposed four-step surface preparation protocol.

**Figure 4 polymers-16-02171-f004:**
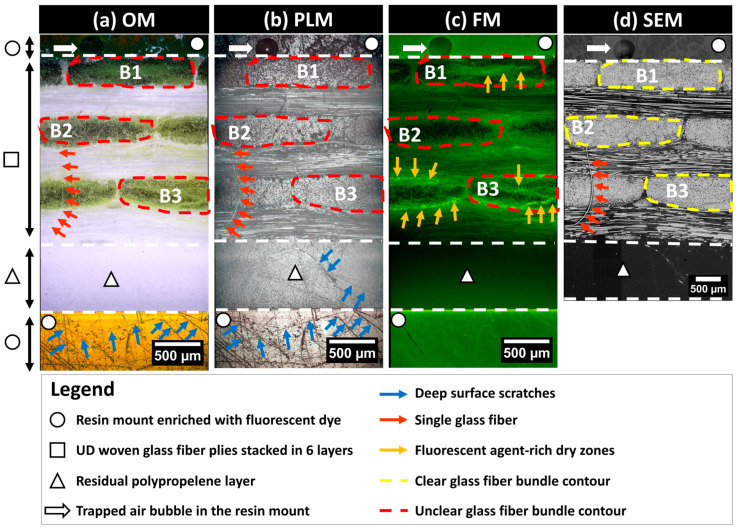
Images obtained using optical microscopy (OM), polarized light microscopy (PLM), fluorescence microscopy (FM), and scanning electron microscopy (SEM) of the same surface polished to 4000 grit following the application of the first fluorescent dye-enriched resin mount.

**Figure 5 polymers-16-02171-f005:**
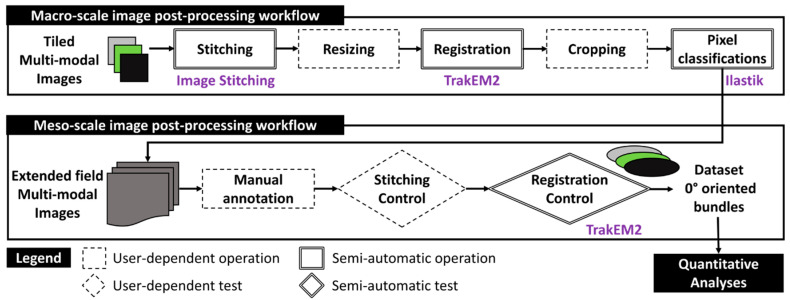
Schematic representation of the post-processing workflows at the macro- and meso-scales for generating bundle datasets from input tiled multimodal images.

**Figure 6 polymers-16-02171-f006:**
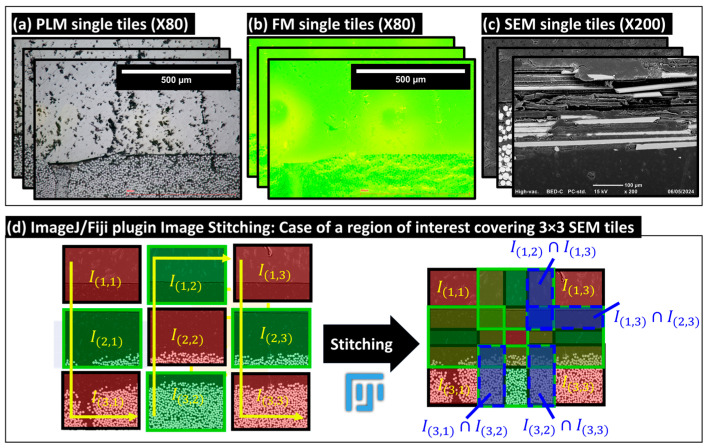
Illustrative examples of collected tiled images: (**a**) polarized light microscopy (PLM), (**b**) fluorescence microscopy (FM), and (**c**) scanning electron microscopy (SEM). (**d**) Example of the stitching operation for a grid of 3 × 3 SEM local images, demonstrating the creation of an extended-field image from individual tiles.

**Figure 7 polymers-16-02171-f007:**
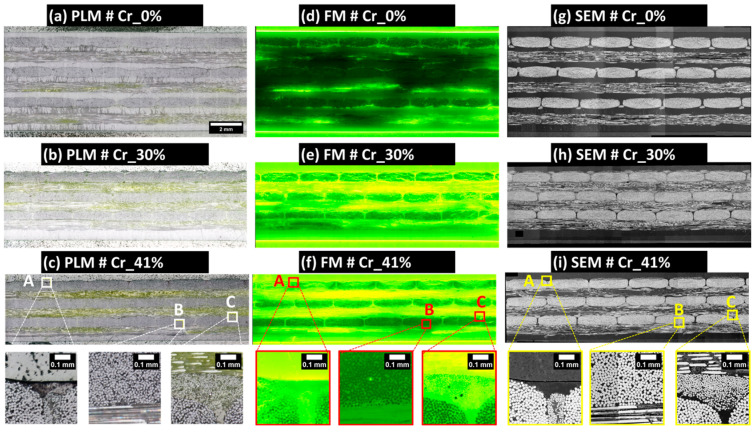
Overview of the final extended-field images of the samples generated using an image processing workflow based on the open-source software ImageJ/Fiji. The images underwent stitching, resizing, registration, and cropping operations. All extended-field images in (**a**–**i**) conform to the scale bar indicated in (**a**). Localized details illustrating full-scale are highlighted in zones A, B, and C. Images (**a**–**c**) were obtained using PLM, images (**d**–**f**) using FM, and images (**g**–**i**) using SEM.

**Figure 8 polymers-16-02171-f008:**
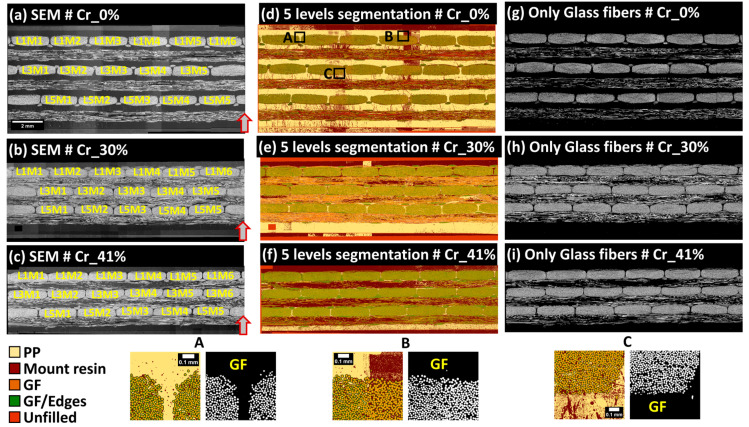
Final extended field SEM images of samples Cr_0%, Cr_30%, and Cr_41% are shown in (**a**–**c**), respectively, alongside five-label pixel classification based on Illastik (version 1.3.3) in (**d**–**f**) and segmented glass fiber pixels in (**g**–**i**). Bundle nomenclatures are based on SEM images, with glass fibers extracted using a five-label Random Forest pixel classification. Illustrative classifications of pixels in zones A, B, and C are provided with corresponding GF segmentations. Arrows denote the orientation of the plate’s thickness within the manufacturing mold. The similarity in nomenclature designation (LiMj) of bundles does not consider any bundle order correspondence between the plates. All extended-field images in (**a**–**i**) conform to the scale bar indicated in (**a**).

**Figure 9 polymers-16-02171-f009:**
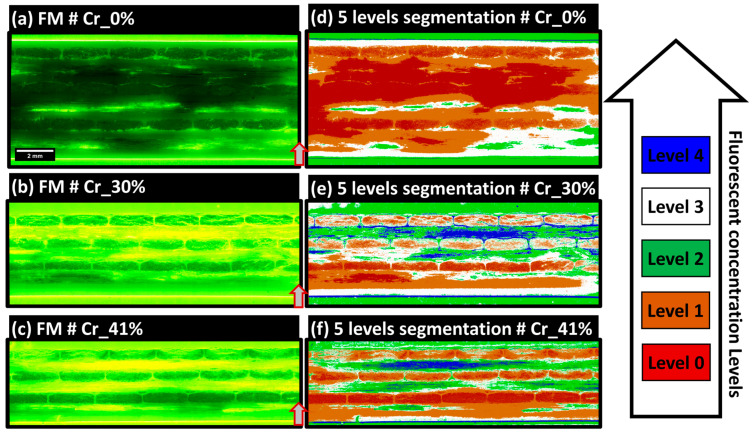
Final extended-field FM images of samples Cr_0%, Cr_30%, and Cr_41% are shown in (**a**–**c**) respectively, accompanied by segmented concentration level domains in (**d**–**f**) respectively. Arrows indicate the orientation of the plate’s thickness within the manufacturing mold. All extended-field images in (**a**–**f**) conform to the scale bar indicated in (**a**).

**Figure 10 polymers-16-02171-f010:**
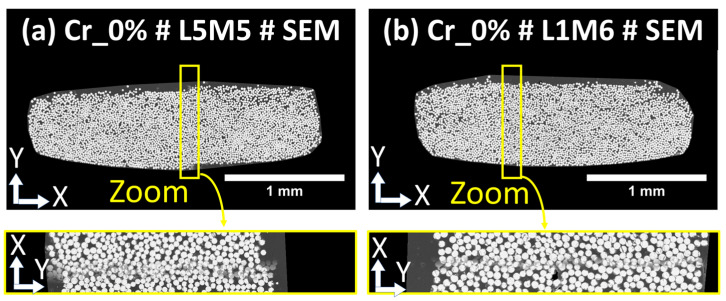
Examples of misalignment detected locally after the macro-scale automatic stitching of the single tiles. (**a**) corresponds to bundle L5M5 from the Cr_0% sample, and (**b**) corresponds to bundle L1M6 from the Cr_0% sample. Rectangles indicate zones of limited stitching precision, marked by shadows from non-overlapping single glass fibers due to the linear blending of misaligned adjacent image tiles.

**Figure 11 polymers-16-02171-f011:**
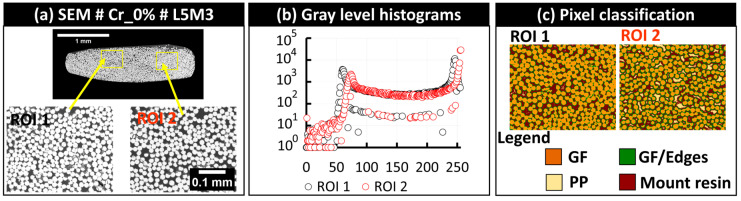
(**a**) Effect of local brightness in SEM images shown through two regions of interest, (**b**) the corresponding grayscale histograms, and (**c**) the corresponding Random Forest pixel classification output.

**Figure 12 polymers-16-02171-f012:**
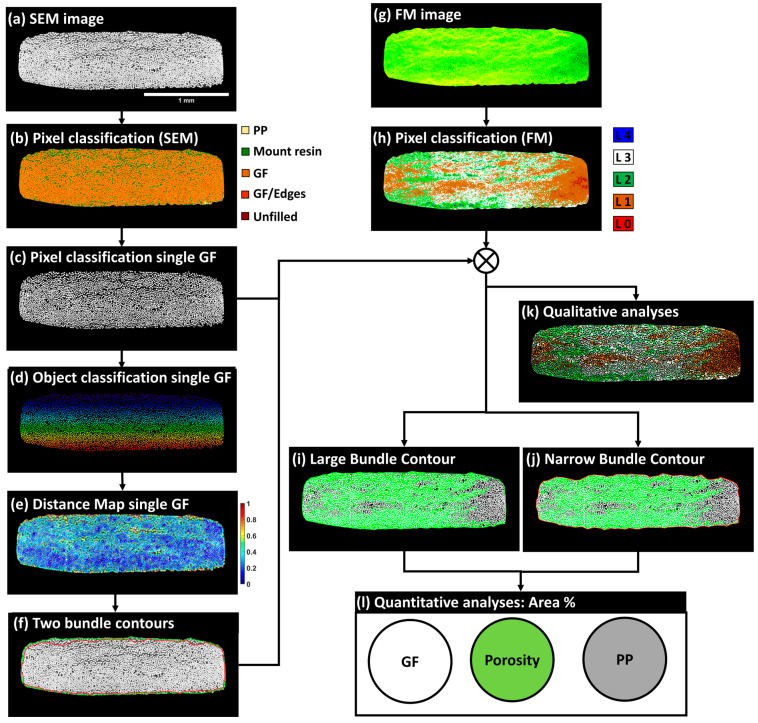
Suggested imaging workflow at the meso-scale of fiber bundles for the quantification of area fractions of GF, PP, and porosity. (**a**) SEM image of bundle L1M1 from sample CR_41%, (**b**) Pixel classification of the SEM image, (**c**) Segmented GF pixels, (**d**) Object classification of a single GF, (**e**) Distance map of a single GF, (**f**) Two bundle contours: the contour in green represents a large bundle, and the contour in red represents a tight bundle, (**g**) FM image of bundle L1M1 from sample CR_41%, (**h**) Pixel classification of the FM image, (**i**) Contour of the large bundle, (**j**) Contour of the narrow bundle, (**k**) Synthetic image based on segmented glass fibers of SEM image and classified pixels of the FM image, (**l**) Quantitative analysis of the area percentage of GF, PP, and porosity.

**Figure 13 polymers-16-02171-f013:**
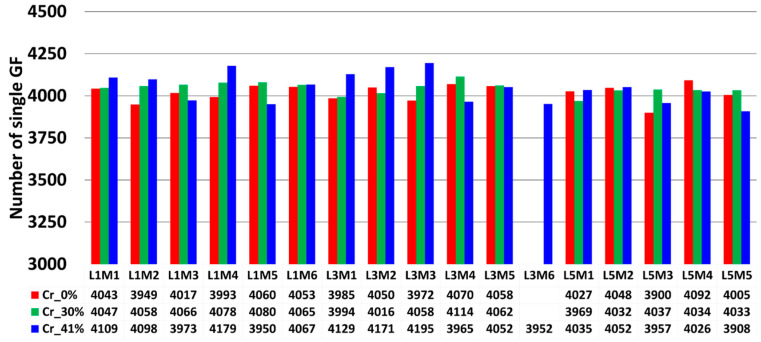
Overview of the total extracted single glass fiber filaments from all non-excluded 0°-oriented fiber bundles. The color indicates a change in the considered composite sample. The similarity in nomenclature designation (LiMj) of bundles does not consider any bundle order correspondence between the plates.

**Figure 14 polymers-16-02171-f014:**
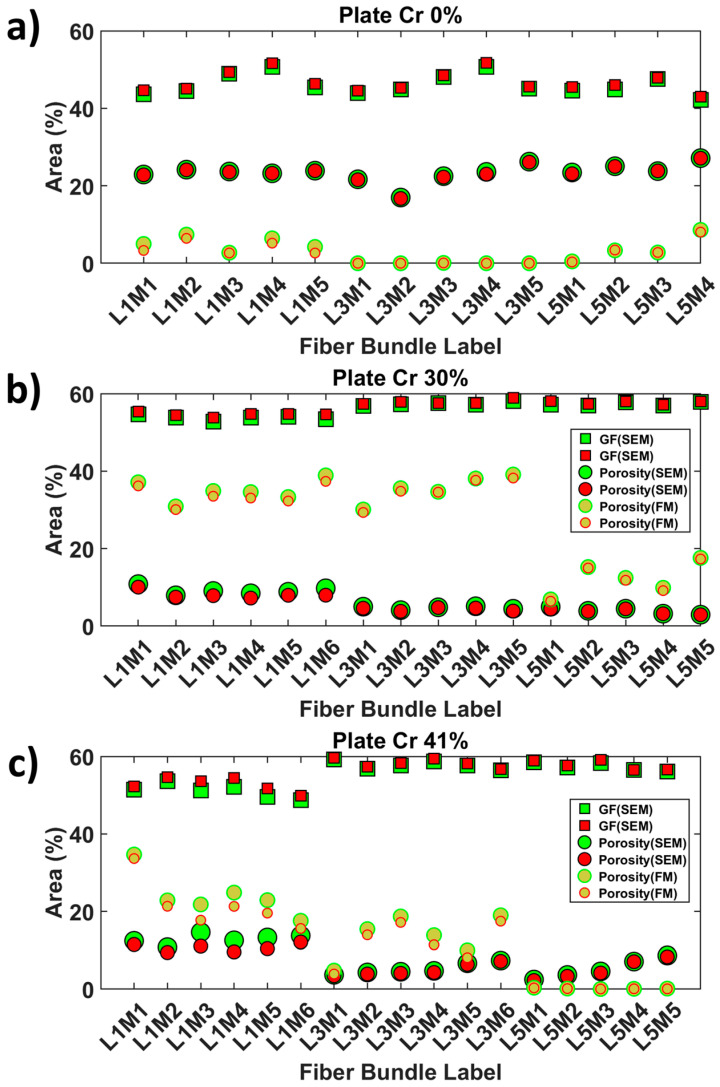
Graphical representation of the meso-scale quantitative output of the area fractions of porosity and glass fibers within 0°-oriented fiber bundles, considering the narrow (red bundle contour) and large (green bundle contour) bundle contours. GF and porosity area percentages in SEM and FM images: (**a**) Cr_0%, (**b**) Cr_30%, and (**c**) Cr_41% samples.

**Table 1 polymers-16-02171-t001:** Overview of microscopy-based imaging techniques for characterizing polymer matrix composites focused on references [3,4,5,6,7,8,9,10,11,12,13,14]. For each reference, the (X) mark indicates the considered imaging technique, length scale, and focus of the analysis. Shaded cells indicate the non-considered aspects.

Ref.	Techniques Used				Scale			Analysis Focus		
	OM	SEM	FM	Other	Micro	Meso	Macro	Porosity	Bundles	Impregnation
[3]	X	X			X			X		
[4]	X				X	X		X		
[5]	X			X				X		
[6]	X					X	X	X		
[7]		X		X	X	X	X	X		X
[8]		X		X				X		
[9]	X					X	X		X	
[10]	X	X			X	X	X		X	
[11]		X						X		
[12]	X	X			X	X				X
[13]	X	X		X						
[14]	X							X		

**Table 2 polymers-16-02171-t002:** Control parameters before and after the manufacturing of the composite plates. V_f_ refers to the measured final fiber volume fractions after manufacturing, while V_f_* refers to the targeted fiber volume fractions set before manufacturing the composite plates.

	Manufacturing					Metrological Control				Microscopy Control				Burn-Off Test	
Plate	Lay-Up	Config.	Weight (g)		V_f_* (%)	Thickness (mm)		Cr (%)	V_f_ (%)	Thickness (mm)		Cr (%)	V_f_ (%)	V_f_ (%)	
			Initial	Final		Avg	StDev			Avg	StDev			Avg	StDev
**Cr_0%**	[0/90]_3_	Film Stacking	1397	1350	42.2	6.1	0.08	0	41.6	6.2	0.01	0	40.0	38.6	0.1
**Cr_30%**		Simplified-CRTM	1408	1081	45.2	4.2	0.13	30.7	60.1	4.4	0.01	30.2	57.4	58.4	0.9
**Cr_41%**		Simplified-CRTM	1392	1037	63.2	3.5	0.08	41.9	71.7	3.7	0.09	40.9	67.8	64.9	0.4

**Table 3 polymers-16-02171-t003:** One-way ANOVA results for the degree of impregnation of 0°-oriented fiber bundles across different manufacturing configurations. The table presents the average degree of impregnation for each layer, the manufacturing configurations, and the corresponding *p*-values for both large and narrow contour data.

	Degree of Impregnation (%)					
	Narrow Contour			Large Contour		
Manufacturing Conditions	Layer 1 (%)	Layer 3 (%)	Layer 5 (%)	Layer 1 (%)	Layer 3 (%)	Layer 5 (%)
**Cr_0%**	90.34 ± 3.46	99.95 ± 0.07	93.23 ± 5.86	92.28 ±3.23	99.95 ±0.06	93.39 ± 5.58
**Cr_30%**	24.44 ± 6.14	16.66 ± 8.91	70.88 ± 10.10	25.60 ± 6.23	17.08 ± 9.2	71.77 ± 10.38
**Cr_41%**	50.51 ± 12.28	67.95± 12.50	99.67 ± 0.31	54.32 ± 13.49	71.41 ± 12.31	99.67 ± 0.26
***p*-value**						
**Hypothesis 1**	0.0530			0.0463		
**Hypothesis 2**	0	0	0	0	0	0

## Data Availability

The datasets for this manuscript are the property of IMT Nord Europe and are not publicly available. Requests to access the primary data should be addressed to the corresponding author. The authors are willing to share the data in the form of an open access article.

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
