# Peer review of "Assessing Intra-Bundle Impregnation in Partially Impregnated Glass Fiber-Reinforced Polypropylene Composites Using a 2D Extended-Field and Multimodal Imaging Approach"

_polymers, 2024, doi:10.3390/polym16152171_

Round 1

Reviewer 1 Report

Comments and Suggestions for Authors

This manuscript presents a detailed methodology for multimodal microscope image analysis of partially-impregnated composites. The methodology is applied to a GF/PP system that has been manufactured by two different means, with results providing clear and interesting trends in through-thickness impregnation. Generally, the quality of the writing and presentation are good, but could be improved in places (see specific comments below). The methodology is of general interest to the composites manufacturing community, but claims of the work’s novelty are weakened by a bibliometric analysis that has missed relevant literature on a similar topic. This needs to be addressed and revised, including relevant sections in the introduction (lines 99 – 104). The uncertainty in porosity values due to brightness change effects should also be quantified and discussed due to its importance in the context of the manuscript. If addressed, along with other minor issues outlined below, the manuscript is acceptable for publication.

Abstract

Line 11 – “the impregnation degree of intra fiber bundles”.

Line 13 – “compression moulding of film stacks”?

Line 16 – “at fibre-scale resolution”.

Further grammatical errors after this, but I will only refer to those that create significant ambiguity or miscommunication.

1. Introduction

Line 38 – “displacement to the preform”.

Line 48 – “Transverse” is usually used to describe in-plane flow perpendicular to the longitudinal direction. It can be similar to through-thickness flow, but is different. Better to refer to it as through-thickness flow or “out-of-plane” flow so as not to confuse the reader. Please change throughout the manuscript.

Line 55 – If the liquid zone is below the preform, the latter presents its own gravitational force, no?

Line 56 – I’m pretty sure the authors are saying that their simplified-CRTM process assumes isothermal conditions, but I have to infer it from this sentence. Please be explicit and clear in stating assumptions.

Line 61 – If the problem involves through-thickness flow only, why is it necessary to capture anisotropy?

Table 1 – It’s not clear how this table is to be interpreted as no legend is provided – what do the question marks and diagonal lines mean? I can guess, but it needs to be clearer. The second box from the bottom right is also blank. Was this intended to be this way?

Line 84 – The authors introduce μCT as an acronym without giving it its full name.

Line 125 – The authors state that the preform is symmetric, but the illustration in Fig 2 and micrographs in Fig 3 show that it is actually asymmetric about the mid-plane. Please correct throughout the text or clarify what you mean by “symmetric”, as convention typically refers to symmetry about the mid-plane of the composite laminate.

2. Bibliometric analysis

This section is interesting if not somewhat out of place and a bit of a non-sequitur. Accepting that is “non-extensive” analysis, I feel that the method should still have been tested. After all, the choice and combination of keywords are open to human interpretation. Having done a quick Google search, I found a conference paper on dual modality imaging by Kim et al. (ICCM18, 2011), one journal paper featuring multimodal imaging for damage detection and characterisation by Shoukroun et al. (Composites Part B, 2020), and a second journal paper using multimodal imaging to inspect resin-rich areas and fibre orientation by Glinz et al. (Journal of Materials Science, 2021). This would suggest serious limitations to the analysis performed here. Please revise this section as well as the relevant section in the introduction to reflect this, whilst also discussing any relevant literature on multimodal imaging of composites.

3. Materials and methods

Line 186 and 187 – The sentence in the parentheses reads poorly. I recommend changing to something clearer (i.e. with different degrees of impregnation).

Figure 2 – The meaning of Cr should be provided in the caption.

Table 2 – The meaning of Vf* should be provided in the caption. Also, the authors refer here to calcination, but burn off in the main text. For the sake of consistency and convention, I would stick with “burn off” in the table too.

Line 380 – 384 – If the polishing parameters are useful for replicating the process, then it would be worthwhile to share them with readers – they could be included in the appendix.

Line 399 – 405 – Split this into three sentences.

Line 409 – “jamping”?

Figure 5 – Incomplete annotations i.e. arrows without words to explain what they are pointing to. In lines 444 – 464, the authors do a good job of highlighting the features that each technique can capture. This should be reflected in the annotations of Figure 5 to provide a clearer message to the reader e.g. complete annotations pointing to the scratches in the PLM image, complete annotations highlighting the clarity of tows and individual fibres for SEM, and complete annotations showing dry spots in the FM image.

Line 469 – “analyses” or “acquisitions”, not both. Alternatively, “Full-scale, extended field acquisitions and analyses of images”.

Table 3 – “Plate” should be in bold for consistency with the other column headings. Also, I’m not convinced this table needs to be in the main text. I think it could be moved to the appendices.

4. Post-processing multimodal images

Line 547 – “similarity in acquiring the grids of tile images”

Line 585 – “Random”

Lines 729 – 731 – Is the distinction between PP and mounting epoxy not significant for accurate determination of porosity, considering that the epoxy will likely infiltrate pores? The authors mention that the GF content is used to estimate the uncertainty of the brightness change effects, but it seems it should be the porosity that should be used to estimate the uncertainty. Please comment.

5. Results

Section 5.2 – I think it would be worthwhile to discuss the results in the context of fibre volume fraction (i.e., with reference to Table 2 and the values given there).

References

Several are incomplete, e.g., references 5, 8, 11, and 12 are all missing the journal name

Comments on the Quality of English Language

Generally, the quality of the writing and presentation are good, but could be improved in places (see specific comments under "Comments and Suggestions for Authors
").

Author Response

Please check the detailed responses included in the attached .pdf file (author-coverletter-38243694.v3.pdf).

Reviewer 2 Report

Comments and Suggestions for Authors

The work is devoted to the application of a multimodal approach based on the study of fiberglass samples using 3 methods of surface microscopy. The results of the work are clearly described using image processing methods and statistical analysis methods. Despite the high level for the manuscript, there are some critical comments that would improve the overall understanding and readability of the article.

1. It would be better to remove the sentence "...multimodal microscopy..." from the title of the article and replace it with, for example, "multimodal imaging approach". You know that multimodal microscopy is a different type of optical microscopy/tomography (Multimodal optical coherence microscopy).

2. The text of the article is very long and some sections can be moved to supplementary materials or deleted.

In particular, Section "2. Bibliometric analysis: Verification of literature gap assumptions". It does not carry scientific information. This is a classic routine search that is carried out before writing any article. Please, move it to Appendix S1.

3. Section 3. page 5 lines 181-193. the paragraph is duplicate the description written at the end of the introduction, lines 131-136. This paragraph can be deleted.

4. Description of the preparation of polished sections for microscopy is a routine methodology that also does not require such a detailed description. Please give a short description of 3-5 sentences in sections "3.2.1. Fabrication of composite plates"  with tabel 2 and "3.2.2. Mechanical polishing of partially-impregnated composite samples". Please,  move the text of these sections to the supplementary materials.

5. Page 12, Figure 5. The scale bar in all images is the same, but the magnification in the images is different. This causes some dissonance. Please remove the magnification values ​​in this figure, they are not needed here, the scale bar is sufficient.

6. on the other hand, some figures do not have a scale bar. Please?check it . If the scale is the same, one sale bar will be enough for all images.

7. Page 12, Section "3.2.3. Full-scale and extended field analyses acquisitions of images" is not of scientific significance, it can also be moved to the supplementary materials or deleted.

8. Section "4.1. Macro-scale stitching: reconstruction of extended-field and full-scale images". The methods of image formation described in this section are also classical and do not require detailed explanation. This section can also be moved to supplementary materials.

Thus,  the perception and readability of the article will improve by shortening the main text of the manuscript. do not forgot that  the main idea of ​​the article is not in the methods of sample polishing, image resolution (number of pixels) or images stitching, but in the complex analysis of images by various techniques and the identification of the porosity of composites.

Author Response

Please check the detailed responses included in the attached .pdf file (author-coverletter-38243894.v2.pdf).

Round 2

Reviewer 2 Report

Comments and Suggestions for Authors

The authors have substantially reorganized the manuscript. The manuscript is recommended for acceptance for publication.